# Study on Detection of a Small Magnetic Particle Using Thin Film Magneto-Impedance Sensor with Subjecting to Strong Normal Field

**DOI:** 10.3390/mi13081199

**Published:** 2022-07-28

**Authors:** Tomoo Nakai

**Affiliations:** Industrial Technology Institute, Miyagi Prefectural Government, Sendai 981-3206, Japan; nakai-to693@pref.miyagi.lg.jp; Tel.: +81-22-377-8700

**Keywords:** thin film, magnetic field sensor, magneto-impedance, small particle, detection of contamination, high-frequency, normal field

## Abstract

This paper deals with the detection of small magnetization using a thin film magneto-impedance sensor with subjecting to strong normal field. The sensor was made by soft magnetic amorphous thin-film with uniaxial magnetic anisotropy in the width direction of the element. It was reported that the sensor has very high sensitivity, such as pico-tesla order, when it is driven by hundreds of MHz. In this paper, a sensitive measurement method aiming for detection of a small particle or a cluster of nano-particles, having low-remanence, is proposed. The point is the application of strong normal field in the measurement area including sensor element and particle. The normal strong field is applied in the normal direction of the sensor plane in the value almost hundreds of mT. Instead of such strong normal field, the sensor keeps high sensitivity, because of the demagnetizing force in the thickness direction. A theoretical estimation for clarifying an efficiency of the method, experimental results of sensor property and sensitivity with subjecting to the normal field, and also a confirmation of detection of a small particle using the proposed method is reported. As a special mention, detection fundamentals when a applied surface normal field has a distribution and also a particle would run through in the vicinity of sensor is discussed.

## 1. Introduction

Detection of small magnetic particle is important for avoiding a contamination of industrial and chemical products. It is also important for detecting magnetic nano-particles included in cells and biomolecules for medical and healthcare technology [1].

For industrial cases, inspection of all items in manufacturing processes is desirable for the recent advanced manufacturing systems, aiming at a reduction of product’s defects, a detection of damaged machine tools, and a tuning of the processing conditions. For example, the detection of small conductive particle is important for avoiding a contamination of high-capacity battery, such as Li-ion secondary batteries. The inclusion of conductive particle in the insulation layer, which is called “separator” of Li-ion battery, causes an extraordinary heating while operation. Our pre-liminary study for the magnetic property of small particle, which was actually sampled from raw material of Li-ion battery, shows that it contains a certain amount of low-remanence magnetic particle. The particle would be assumed to come from a naturally included in graphite minerals and also tool steel chips of grinding machines. This type of battery is getting used in the field of air-plane power-supply or life support robots. The extraordinary heating or unexpected shutdown of the system has a possibility to make a disaster. Even if a very rare inclusion of such small particle, it would be recognized as a serious problem. Therefore, full inspection of small particle, such as under one hundred micro-meters diameter, while fabrication process is needed for Li-ion separator. The detection must be applicable for a kind of roll-to-roll method with the width of the separator sheet more than 1 m and the running speed more than 100 m/min. Another example of detecting a magnetic small piece is a detection of metal contaminant in Aluminum material or product. Based on recent needs for recycling natural resources, Aluminum recycled metal is widely used in industrial products, such as an automobile engine. An inclusion of iron chips in Aluminum products have a certain possibility of inducing tool brakeage in post-process, such as drilling. It also has a possibility of tearing a sealing resin, when the contaminant chip appears on the surface of seal flange.

A recent progress in medical technology utilizes magnetic nano-particles to be able to introduce in a human body. The Resovist^®^, a kind of Super Paramagnetic Iron Oxide (SPIO) nano-particle, is well known that it is used for a contrast-enhancement of the Magnetic Resonance Imaging (MRI). It consists of a mixture of Fe_2_O_3_ and Fe_3_O_4_ nano-particles having a paramagnetic property. It has a bio-compatibility against human body, therefore many trials of medical application using Iron Oxide magnetic nano-particles have been studied recently, such as a magnetic separation [2], a magnetic drug delivery [3], a magnetic targeting [4], and a hyperthermia for cancer treatment [5]. A detection of locally concentrated magnetic nano-particles such as in a cancer or in a lymph node is getting to be important for the purpose of a preoperative detection of cancer metastasis [6]. A high sensitivity magnetic field sensor with combination of a field generator is a candidate of applying to such a medical inspection.

The giant magneto-impedance effect and sensor is recently attracted huge attention for the field of biosensing, healthcare, and industrial monitoring application, due to an easiness of harnessing it in industries as an actual system. The complicated measurement protocols, expensive equipment, and requisite for cryogenic temperatures for the other sensitive magnetic sensors, such as nuclear magnetic resonance and superconducting quantum interference, have prevented them from adopting it in industries.

The senor system, which is proposed in this paper, consists of a combination of thin film magneto-impedance (MI) sensor [7] and high-frequency detection circuit over hundreds of MHz [8]. The use of high-frequency makes the MI sensor to be high-sensitivity, due to a thin skin depth. In order to detect such particles effectively, we propose a method which is in combination with high-sensitivity sensor made by thin film and applying a strong magnetic field in the measurement area [9,10]. The strong magnetic field is applied for the purpose of magnetize the low-remanece particle stronger. Instead of such strong field, applied in the direction of the surface normal direction against the sensor substrate plane, the thin film sensor keeps high sensitivity, because of the demagnetizing force in the thickness direction. The point of this study is the application of strong static normal field in the measurement area including both sensor element and adjacent small particle. The normal field would better be made by hard-magnet in order to decrease energy consumption [11]. The magnetic contaminant of natural resources and also a tool steel chip of industrial processing has a tendency of low-remanence. The Resovist^®^, which consists of a mixture of Fe_2_O_3_ and Fe_3_O_4_ nano-particles, is well-known that it has a paramagnetic property, then, a cluster of this nano-particle has also low-remanence property.

Recent study on sensing a micrometer-sized magnetic bead or nano-particle was for detection or identification for biomolecules. Some of them tried to detect a certain amount of deposited beads on sensor surface [12] or density of dispersive solution of beads [13]. Various sensors such as giant magneto-resistive (GMR), spin valve, giant magneto-impedance (GMI), were report-ed [14,15,16,17,18,19,20]. The bead was usually low-remanence, so some of them used a method of application of alternating normal field with combination of lock-in technique such as 200 Hz–121 Oe [7] and 200 Hz–50 Oe [14]. But it is obvious that the stronger normal field makes the particle magnetization larger and then the sensitivity higher. The application of more than 100 mT (1000 Oe) of static surface normal field to the high-sensitive thin-film GMI sensor and also apply it to industrial fabrication process and biomedical application for the purpose of detecting moving small magnetic particle is our proposal.

In this study, a method of detection of small magnetic particle, which has low-remanence property, using a thin film magneto-impedance sensor is investigated. The point of this study is the application of strong static normal field in the measurement area including both sensor element and small particle. This paper composed of a theoretical estimation for clarifying efficiency of the proposed detection method, an experimental measurement of sensor property and sensitivity of the sensing system, while exposed in the surface normal field, and a confirmation of detection of small particle using this method, especially in the vicinity of sensor element. This paper shows a comprehensive fundamentals of magnetic particle detection using thin film MI sensor operated in a strong surface normal magnetic field.

This paper is constructed based on a Japanese paper of technical meeting [21], which was copyrighted 2016 IEEJ, and Japanese patent [22] applied in 2013, to be rewritten as an English article with some important additional data and discussion.

## 2. Theoretical Formulation of Sensing a Vertically Magnetized Particle

### 2.1. Case of Far Distance batween Sensor and Magnetized Particle

Firstly, in order to make clear the efficiency of the proposed method, a theoretical analysis is carried out. Figure 1 shows a schematic illustration of the proposed measurement system. The strong magnetic field in surface normal direction, which is relative to the sensor substrate surface, magnetizes the magnetic particle, consequently its magnetization makes a field in the X direction at the sensor position, which is a measurement direction of the sensor and which is along the sensor plane. Based on the formula of magnetic dipole (1), an estimation of the field is carried out.
(1)$$H=-\frac{1}{4\pi \mu_{0}}\nabla \frac{m\cdot r}{r^{3}}$$
where *m* is the magnetic moment, *r* is the vector of particle position.

For simplify the equation, it is restricted in X-Z plane. Figure 2 shows the schematic of the analysis coordinates. The magnetic dipole moment, approximated as a non-volume point, is placed on the origin of coordinate. A magnetic field in the X direction on the line of *z* = const. is shown in Equation (2).
(2)$$B_{x}=\mu_{0}H_{x}=-\frac{3m}{4\pi }xz{\left(x^{2}+z^{2}\right)}^{-\frac{5}{2}}$$

It is supposed that the magnetization in the volume of particle is a constant value. Then,
(3)$$m\;\left[\mathrm{Wb}\cdot m\right]=I\;\left[\mathrm{Wb}/m^{2}\right]\cdot V\;\left[m^{3}\right]$$
where *I* is the magnetization, *V* is the volume of particle.

Figure 3 shows a variation of magnetic flux density *B_x_* as a function of X position. In this case *z* = 1 mm, particle diameter is 65 μm, and magnetization *I* = 0.2 T. The profile has one maximum and one minimum point and each point is on *x* = ±*z*/2, which is obtained from *dB_x_/dx* = 0.

Figure 4 shows a variation of *B_x_* as a function of *z*, in case of particle diameter 50 μm. The magnetization of the particle is 0.1 T, 0.4 T, and 1.0 T respectively. From this result, the sensor of 10^−9^ T sensitivity can detect the particle with 50 μm of diameter and 0.1 T of magnetization within the distance of *z* = 7.5 mm. From Equation (2) the *B_x_* is proportional to magnetization *I*.

Figure 5 shows a variation of *B_x_* as a function of *z*, in case of particle magnetization 1.0 T. The diameter of the particle is 20 μm, 65 μm, and 200 μm respectively. From this result, the sensor of 10^−9^ T sensitivity can detect the particle with 20 μm of diameter and 1.0 T of magnetization, within the distance of *z* = 6.5 mm. From Equations (2) and (3) the *B_x_* is proportional to the cube of the particle diameter.

These results show, in order to detect a tens of micrometer magnetic particle by a sensitive sensor, the particle must be magnetized almost saturation, that is why application of strong normal field is needed, and placed it in the vicinity of the sensor.

### 2.2. Case of a Magnetized Particle Running in the Vicinity of Sensor Element

The previous subsection described a case in which the magnetic field at the sensor position was uniform and independent of sensor element position. This situation arises when the distance between the sensor and the magnetized particle is large enough.

In this subsection, a discussion is made for the case when a magnetized particle is running in the vicinity of sensor element.

Figure 6 explains an interaction between the magnetized particle and thin film sensor in a vertical magnetic field. The magnetic particle is magnetized in the vertical direction just above the sensor element, and the generated field affected the sensor output caused by a positional changing magnetic field. An effect of a distributed magnetic field on a sensor impedance is formulated as follows;

The high-frequency impedance of the magneto-impedance sensor is well-known that it is caused by the skin-effect in the sensor element. The skin-depth of the current flow changes as a function of high-frequency permeability of the soft magnetic sensor element. In case of a frequency range which is larger than roughly MHz, the permeability appears based on a magnetization vibration, instead of the magnetic domain wall movement.

Figure 7 indicates a schematic explanation of magnetization vibration in the thin film magneto-impedance sensor, when a surface normal field is applied. The left three figures show a variation of magnetic moment, in which the orientation and the vibration are in each cases of different external magnetic field along the sensor length direction. The upper right schematic explains a moment vibration as a perturbation around a stable state. The right sectional schematic explains the effect of demagnetizing force along the thickness direction. In this figure, the magnetic anisotropy field is shown as *H*_k_. The direction of easy axis of the magnetic anisotropy lies in width direction in this Figure. Based on our study, the domain variation depends on the easy-axis direction. In case of the easy-axis directing along the in-plane inclined direction, a domain wall movement appears and the area ratio of the contiguous domain changes as a function of applied field. The applied field is in the direction of element’s longitudinal axis. The left schematic of Figure 7 is the case which has the easy-axis in an ideal width direction. The domain width in our Co_85_Nb_12_Zr_3_ element is roughly 20 μm to 50 μm which is narrow enough compared with the whole element length, 1000 μm to 2000 μm, then a positional variation of magnetic field, which is applied in the sensing direction, has a possibility to effect on the partial element impedance proportionally changing to the external field within a continuous strip element.

Based on the assumption mentioned above, the whole impedance of the sensor element is assumed to estimate as an integral of partial impedance of the element as shown in Figure 8. In case of a *L* mm length sensor, which is placed from *x* = 0 to *x* = *L*, the total impedance is expressed as Equation (4)
(4)$$Z_{total}\left(H_{x}\right)=\int_{0}^{L}Z\left(H_{x},x\right)dx$$

Here, $H_{x}$ is a sensing magnetic field at the position *x*.

The following experimental results and discussions in this paper are constructed based on this equation.

## 3. Experiment and Discussion

This section reported an effect of magnetic field variation on a sensor impedance, when a strip of thin film magneto-impedance sensor is placed in a surface normal magnetic field and detects a vertically magnetized magnetic small particle in the vicinity of the sensor. This section discusses about two viewpoints. The former explains an effect of distribution of the normal magnetic field for the sensitivity, and the latter reports and discusses a result of actual particle detection running in the vicinity of the sensor element.

### 3.1. Sensor Impedance Property and Sensitivity

#### 3.1.1. Variation of MI-Curve as a Function of Normal Field

Variation of the MI-curve and the sensitivity of the sensor are experimentally measured as a parameter of the normal field. The MI-curve means the variation of sensor impedance as a function of external magnetic field applied along the in-plane sensing direction, which is the X direction in Figure 1.

The sensor element was fabricated by a thin film process. An amorphous Co_85_Nb_12_Zr_3_ film was RF-sputter deposited onto soda glass substrate and then micro-fabricated into rectangular elements by lift-off process. The dimensions of the element are ranging from 1000 μm to 2000 μm of length, 20 μm to 50 μm of width, and 1.35 μm to 2.15 μm of thickness. The element was annealed in magnetic field in order to induce uniaxial magnetic anisotropy. The direction of the magnetic anisotropy was controlled by the direction of the magnetic field. In this study, the magnetic field during annealing, 240 kA/m, 673 K, was oriented in short-side axis, therefore width direction, of the element.

Figure 9 is a photograph of sensor element. A coplanar structure was used for fitting a G-S-G type high-frequency wafer probe used for the impedance measurement.

Figure 10 shows a schematic illustration of measurement apparatus. The normal field was generated by two NeFeB magnets, which is fixed on heads of Si steel c-shape core with the opposite poles facing each other, in this study the north pole is placed in the upper side. The sensor element is placed in the middle position between these two magnets. A Helmholtz coil is placed for the purpose of applying magnetic field in sensing direction; X. This apparatus can control the strength of the vertical field by changing the size and the distance of magnets. The sensor impedance was measured by network analyzer using S11 measurement. The frequency of current induced in the sensor was 500 MHz and 800 MHz, and input power was −14 dBm. In this frequency range, a phenomenon which is caused by resonance appears with variation of applied magnetic field. The reason why this frequency was chosen is because an effect of the vertical field for a vibration of magnetic momentum, caused by high-frequency current, can be clearly observed by the experiment.

Figure 11 is a photograph of actual measurement system. A wooden needle was used for feeding a small magnetic particle, with a particle fixing at the tip. The strength of the applied vertical magnetic field was controlled by both the thickness of magnets and the distance of magnets.

Figure 12, Figure 13 and Figure 14 show a variation of sensor impedance as a function of external field in X direction. The applied surface normal field was 0 A/m, 83.2 kA/m and 129 kA/m respectively. Figure 12 represents an impedance variation for 1000 μm length element with 800 MHz of induced current. Figure 13 represents an impedance variation for 1000 (micro-)m with 500 MHz. And Figure 14 represents an impedance variation for 2000 μm length with 500 MHz. In these figures the caption “a” represents a variation of |Z|, the caption “b” represents a variation of real part of impedance, Re(Z), and the caption “c” represents a variation of imaginary part of impedance, Im(Z). The results for 1000 μm length element show that the impedance around zero X-direction fields has slight change even when the surface normal field is changed. Here both the Re(Z) and the Im(Z) has slight change. On the other hand, the peak value of impedance decreases as the normal field increases. The peak value of impedance decreases for the both case of Re(Z) and Im(Z). For the range where the impedance changes rapidly, where the |*H*| is ranging from 5 to 15 Oe (almost 400 A/m to 1200 A/m), the Im(Z) has negative and minimum value for the case of Figure 12, that means an existence of resonance. As increasing the normal field, the absolute value of the minimum Im(Z), i.e. |min Im(Z)|, decreases. On the other hand, the Figure 14 represents different properties. For the case of this 2000 μm element, the peak value decreases as same as previous one, but the impedance value around zero increases and finally has a peak value as the normal field increases.

The magnetic domain of element was Landau-Lifshitz-domain when both the X-direction field and normal field is zero. The magnetization process of the element which has uniaxial easy axis in width direction is magnetization rotation. Figure 7 shows a schematic explanation of arrangement of magnetic momentum in the sensor element as a function of external field. When the X-direction field is zero, the momentum orients in width direction, therefore Y-direction, forming contiguous opposite domain areas. As increasing the X-direction field, the momentum rotates in the X direction according to the strength of applied field with keeping the wall position. In this case, the high-frequency permeability, which has influence on sensor impedance, initially increases, then has a maximum at the applied field nearly equals the anisotropy field, *H*k, and after that decreases as the X-direction field increases. This high-frequency permeability arises from the vibration of momentum which is explained by perturbation theory. From the result of unchanged impedance around zero fields, in spite of the surface normal field application, it is supposed that the normal field, within the value of our experiment, has no effect of enhancement or decline of permeability, therefore it has slight effect for changing the direction of momentum or changing the perturbation potential distribution. On the other hand, the decrement of maximum value means that the normal field has an effect of decline permeability. With consideration of decrement of |min Im(Z)|, the normal field is supposed to have an effect of decline permeability both real and imaginary within the range where the impedance change rapidly until the area around it has maximum. The details of this consideration would be a future subject of this study.

There is another explanation for the change of MI-curve caused by applying normal field. Our measurement apparatus has un-uniformity of X-direction field within the length of the sensor, more than several Oe. This un-uniformity also has the effect of making the impedance profile flat. It is based on Figure 8 and Equation (4). An impedance of whole element of a sensor is assumed to be estimated by an integral of impedance of small section Δ*x*.

Whereas if a normal field with distributed vector in X-direction is applied, the applied field on the sensor element has a distributed profile in X direction (Figure 15). This distributed X-directional field makes a change of MI-profile on individual Δ*x* pieces as shown in Figure 16. As a result, the un-uniformity of X-field makes the MI-curve of whole element to be flat. In this case, the sensor impedance is explained as following equation.
(5)$$Z_{total}\left(H_{x}\right)=\int_{0}^{L}Z\left(H_{x}+h_{bx}\left(x\right),x\right)dx.$$
where applied normal field on the sensor film *H_normal_* is shown as follows;
(6)$$H_{normal}=H_{z}+h_{bx}\left(x\right),\;H_{z}=const.$$

The result in Figure 14 shows a measurement for 2000 μm length element. The difference of variation of MI-curve with changing the normal field, compared with the result of Figure 13 of 1000 μm element, would be affected by the distribution of X-direction field.

Based on these discussions, a generating apparatus of the surface normal magnetic field which can make more uniform in X-direction and stronger in Z-direction and can control them individually is effective for clarify the mechanism of the sensor surface normal field.

#### 3.1.2. Sensitivity Evaluation Using Carrier-Suppressing Circuit

The sensor system in this study is a combination of thin film MI sensor and high-frequency measurement circuit. A well-known combination is with the carrier- suppressing circuit [6]. Figure 17 shows the block diagram of the circuit. The device list is shown in Table 1. A high-frequency alternating signal *f*_0_, which is called “carrier-signal”, is divided into two. One is inputted to the sensor, and the reflection signal from the sensor is introduced to the right side divider. Another signal is set in the same amplitude and opposite phase as the sensor reflection signal. Consequently, the combined signal come out from right divider is a carrier eliminated signal. When the ac magnetic field *f*_ac_ is applied to the sensor, the output signal has spectrum of *f*_0_ ± *f*_ac_, which is known as side-band spectrum. The merit of this circuit is the very low noise level in the side-band frequency, because of the effect of carrier-suppressing. In this study, the sensitivity of the sensor with subjecting to the surface normal field is evaluated by using this circuit. The sensor element is the same one as measured in Figure 12 and Figure 13. The high-frequency carrier signal come out from signal-generator (S.G.) was −8 dBm, 410 MHz. The applied ac magnetic field to the sensor was 310 Hz.

Figure 18 shows a photograph of whole measurement system, and Figure 19 shows an enlarged view of the carrier-suppressing circuit connected with the sensor. The circuit was covered by Aluminum foil while measurement, for the purpose of reducing measurement noise.

Our original proposal in the circuit was an application of high-frequency circulator. It has a merit of reducing number of connected cables, which is to make as 1-cable, then a reduction of connected electrode-pads and a space reduction of connected cable with the thin film small sensor. It also has a demerit of decreasing sensitivity, due to a signal loss of the circulator itself and an existence of leakage carrier-signal, *f*_0_, from port-1 to port-3 and from port-2 to port-1. The leakage signal reduces the sensitivity especially when the sensing AC field, *f*_ac_, has a low frequency. It induces the deviation of frequency between the *f*_0_ and *f*_ac_ to be reduced, then it makes difficult to detect the signal separately. The performance of the driving circuit for low frequency signal would be improved using a logarithmic amplifier IC-chip, which was tried in this paper in Section 3.2.2, and connected to a development of driving circuit which can detect low frequency signal including DC-signal [23].

The high-frequency devices shown in Figure 17 are listed in Table 1. This trial was carried out almost 10 years ago, then a connection of discrete units was adopted in this experiment. A recent sophisticated IC-tips can reduce the whole circuit volume and increase signal stability using the concept of our proposed circuit. It is shown in our following study [9,10,11].

Figure 20a shows the result of evaluating sensitivity when the normal field is zero. Sensor sensitivity is evaluated by the extrapolation method. The sensitivity was 1.4 nT/Hz^1/2^ in this case. Figure 20b shows the sensitivity with subjecting to normal field of 83.2 kA/m, in this case it was 1.7 nT/Hz^1/2^. These results show that the 83.2 kA/m normal field has slight effect of degradation for the sensor sensitivity.

Figure 21 shows a MI-curve and its tangential line at a bias-point of the high-sensitivity measurement. Figure 21a is the one when the normal field is zero, and Figure 21b is when the normal field is 83.2 kA/m. A value of *dZ/dH* at the bias point represents the sensor sensitivity. A comparison of ratios of sensitivity which is shown in Figure 20a vs. Figure 20b, (1/1.4)/(1/1.7) = 1.21, with the ratios of *dZ/dH* in Figure 21a vs. Figure 21b, (21 mΩ/(A/m))/(17 mΩ/(A/m)) = 1.24, shows that these ratios are in good agreement. Based on this result, a higher sensitivity is realized when a sensor is designed to have larger value of *dZ*/*dH*, which is the same design rule as ordinary magneto-impedance sensor.

This work was a basis of the following developments of both the sensor driving circuit using a chip-sized high frequency devices [23] and also the uniform magnetic field generator of surface normal field [10,11], which is applied to a thin film MI sensor used inside a strong magnetic field. A sensitivity confirmation and magnetic domain variation during the sensor operation in stronger field had been carried out and reported in Ref. [9].

### 3.2. Detection of Small Particle in the Vicinity of the Sensor Element

#### 3.2.1. Measurement in Case of Single Strip Sensor

In this experiment, detection of magnetic small particle using a single sensor strip was carried out. The particle, which has 65 μm diameter and 1 T saturation magnetization, was detected by the sensor with subjecting to 83.2 kA/m normal field. M-H loop of the particle is shown in Figure 22. This particle has remanence of 0.05 T, and corecivity of 11 kA/m. Figure 23 shows a schematic illustration of the measurement system of magnetic small particle. A soft magnetic sphere-shaped particle was mechanically scanned above the sensor element. The whole measurement area which contains both the sensor and the particle was within 83.2 kA/m of the surface normal field. Variation of the field *H*_z_ is less than 5% within 2.5 mm from the center of the area. The sensor element is the same one above mentioned, the length is 1 mm.

Figure 24 shows the result. The measurement circuit was based on the carrier-suppressing method. A modification of final stage was an addition of log-amp detection with combination of an offset compensation and a ×100 amplifier. The particle altitude from the sensor plane was 0.5 mm. The particle was scanned along Y = 0. The 65 μm particle made an output variation between ±6 V with the profile having one maximum and one minimum point at around the longitudinal edges of element.

The consideration of the measurement result on Figure 24 is as following. The output profile with one maximum and one minimum at both edges is the common characteristic for measurement of X-directional field produced by vertically magnetized particle. Figure 25 shows a variation of magnetic field in X-direction as a function of X-position. The 65 μm diameter magnetic particle, which is magnetized in *I* = 0.2 T, is placed on *x* = 0, *y* = 0, and *z* = 0.5 mm. The detection of magnetic field was carried out on *X*-axis. This profile is calculated by Equation (2). Based on this X-field distribution, a consideration is made for an estimation of impedance of whole element of the 1 mm length sensor. The magnetic field induced by the magnetized particle is localized and the effect is changed as a function of particle position, which was shown in Figure 6. A distributed magnetic field is made on the sensor element, which is generated in the vicinity of magnetic dipole, and the distributed profile is changed as a function of particle position, which is running through just above the sensor. An estimation of element impedance as a function of particle position is tried as follows;

Impedance variation of the sensor is shown in collinear approximation in the vicinity of bias point. The field on sensor made by 65 μm particle is so small that this assumption in reasonable. Based on the measured sensor property with applying normal field, Figure 21b, it is assumed as follows;
(7)$$Z\left(\Omega \right)=25.3+1.37H_{x}$$

The element impedance is estimated by the following equation. The estimation is based on the same assumption as Equation (4). The element position and dimensions are the same as shown in Figure 23.
(8)$$Z_{total}=\int_{-1}^{0}Z\left(x\right)dx$$

Then
(9)$$Z_{total}=\int_{-1}^{0}\left(25.3+1.37H_{x}\right)dx$$

Here *H*_x_ is analytically obtained based on the equation of magnetic dipole, Equation (2), when a particle is placed in *x*’(10)$$H_{x}\left(x,x^{\prime }\right)=-\frac{3m}{4\pi \mu_{0}}\left(x-x^{\prime }\right)z\left\{{\left({\left(x-x^{\prime }\right)}^{2}+z^{2}\right)}^{-\frac{5}{2}}\right\}$$.

Then (11)$$Z_{\mathrm{total}\;}\left(x^{\prime }\right)=\int_{-1}^{0}\left(25.3+1.37H_{x}\left(x,x^{\prime }\right)\right)dx$$

A result of numerical estimation is shown in Figure 26. The vertical-axis is an impedance of the element. The output level of measurement system is proportional to the sensor impedance, so Figure 26 shows the reason of the profile having one maximum and one minimum point at around the longitudinal edges of element, shown in Figure 24. The difference between Figure 24 and Figure 26 is assumed that the numerical estimation does not take into consideration of a positional variation of demagnetizing field, which is especially affected in the vicinity of longitudinal edge of the sensor element. It is also assumed that the difference come from an existence of electrode-pads at the sensor edge. There is a certain dimensions of electrode pad, as shown in Figure 9, then an uncertainty of sensor edge appears.

Figure 27 shows a measured variation of impedance as a function of particle position *y* in case of *x* = 0, i.e., the position of impedance peak for Figure 24. The particle height is the same as Figure 24, *z* = 0.5 mm. The application of the surfase normal field makes the measured whole particle magnetize in the same direction Z. This makes the estimation of interaction between the sensor and the particle easy.

#### 3.2.2. Measurement in Case of Meander Shaped Differential Sensor

In this study, a detection of adjacent particle from the sensor element was investigated using differential sensor. The sensor we used was a pair of meander shaped thin film MI sensors, adjacently located on a glass substrate, which is shown in Figure 28. The differential sensor was composed by two meander sensors, each of them consisted of 10 strips of 50 μm width MI sensors. Each strips were connected by Cu connecting strip to form a meander sensor. The typical length of the strips was 1.25 mm, the sensor width was 0.95 mm as a solo one, and the whole width of the differential sensor was 2 mm.

Figure 29 shows a driving circuit of our differential sensor. It consisted of a combination of the previous circuit of this paper, Figure 7, to formed a differential layout. The final detection of this circuit was carried out by a logarithmic amplifier with differential inputs. In this circuit, both the amplitude and the phase of inputted signals were set as the same value using the attenuators and the phase shifters. The logarithmic amplifier can detect 410 MHz signal to a static voltage output which is logarithmically proportional to the differential of two inputs. The output was set as minimum value when a vertical magnetic field was applied and also there was nothing to detect. In this circuit, the maximum output of the final amplifier was 4.3 V, then a saturation appears when the output would be a large value.

Figure 30 is a photograph of measurement system of the differential detection experiment. A G-S-S-G non-magnetic wafer probe was use for contacting the sensor, and a soft magnetic small particle was 2D scanned using XYZ-manipulator. The measurement results are indicated as follows;

Figure 31 is a 2D-mapping of a measured signal, when a particle having a diameter of 200 μm was scanned on the sensor substrate with the height of 1 mm. There is an extreme value just above the right edge of the sensor element, and the polarity was opposite on the differential pair sensor. The measurement was restricted in the left half region, due to the existence of contact needles of wafer probe, but the result is easily estimate on the left end.

For the purpose of confirming the edge effect precisely, measurements of linearly scan along the *X*-axis on different Y-position as a parameter was carried out.

Figure 32 is output profiles as a function of X-position. The Y-position was a parameter. The *y* = 1 and *y* = −1 are the position of outer end strip of the meander sensor. The *y* = 0.5 and *y* = −0.5 are at the middle of each meander sensor. The particle height was 1.5 mm in this case. This result also shows that the extreme value of sensor output appears at the edge of the sensor element, *x* = 0. It is confirmed the validity of the proposed estimation procedure of sensor impedance in case of the magnetic particle is placed in the vicinity of the thin film MI sensor element.

This measurement was carried out using the apparatus shown in Figure 30. In this apparatus, the particle, which was fixed at the tip of wooden needle, was scanned using a manually controlled XYZ-manipulator. The turning of nob was carried out cautiously, but a certain positional uncertainty of the apparatus unit appears, then an asymmetricity occurred on measurement. The output base line of the sensor driving circuit had a slight drifting tendency in this paper, where the experiment was carried out in 2016. It would be improved in later article, such as Ref. [10], based on this paper’s investigation. The asymmetricity of this data, in Figure 32, came from both of these reasons.

## 4. Conclusions

This paper discussed detection fundamentals of magnetic small particle using thin film magneto-impedance sensor. In this study, the particle is magnetized in vertical direction relative to the flat substrate plane of the sensor. There is an application of strong vertical magnetic field in the measurement area including sensor element for the purpose of detecting a low-remanence magnetic particle simultaneously with magnetization. The variation of sensor impedance, which is determined by a magnetic field coming from the particle, was estimated and formulated both the case of far distance and also the case of in the vicinity of sensor element. In case of far distance, the field is estimated using the equation of magnetic dipole. Whereas in the case of adjacent distance, the effect of magnetic field distribution on a sensor strip must be take into consideration using an assumption of impedance integral of partial strips of sensor element. This assumption was experimentally confirmed in two cases. One is the effect of distributed field for the sensor sensitivity. A distribution must be minimized for preventing sensitivity decrement. The other is that it is predicted that the extreme value of sensor impedance would be obtained when a particle is placed just above the sensor longitudinal edge. This phenomenon was confirmed in the both case of single sensor and also differential meander shaped sensor.

This paper is constructed based on a Japanese paper of technical meeting [14] and Japanese patent [15] to be an article of International Journal.

## Figures and Tables

**Figure 1 micromachines-13-01199-f001:**
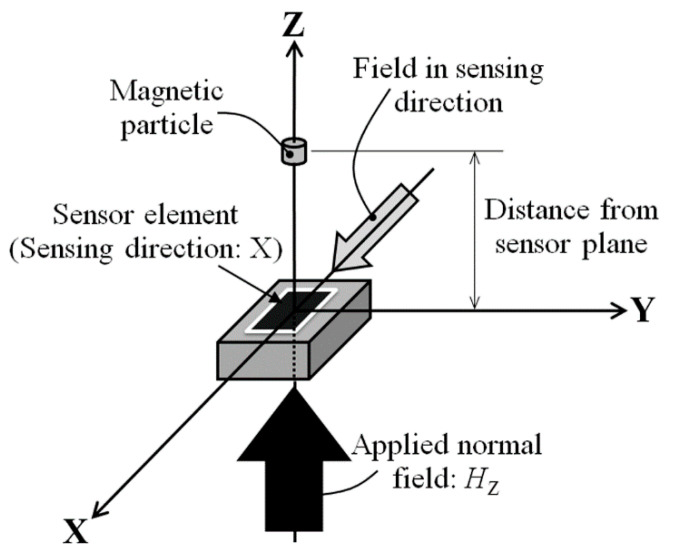
Schematic explanation of measurement system [21].

**Figure 2 micromachines-13-01199-f002:**
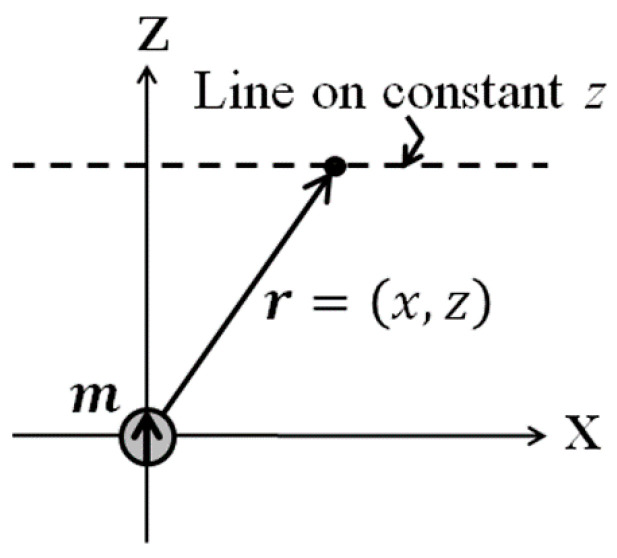
Schematic of the analysis coordinates [21].

**Figure 3 micromachines-13-01199-f003:**
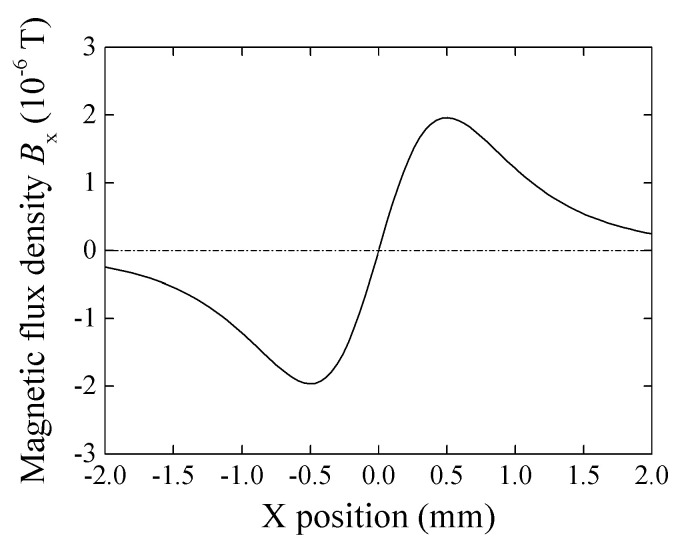
Variation of magnetic flux density in X-direction, *B_x_*, as a function of X-position [21].

**Figure 4 micromachines-13-01199-f004:**
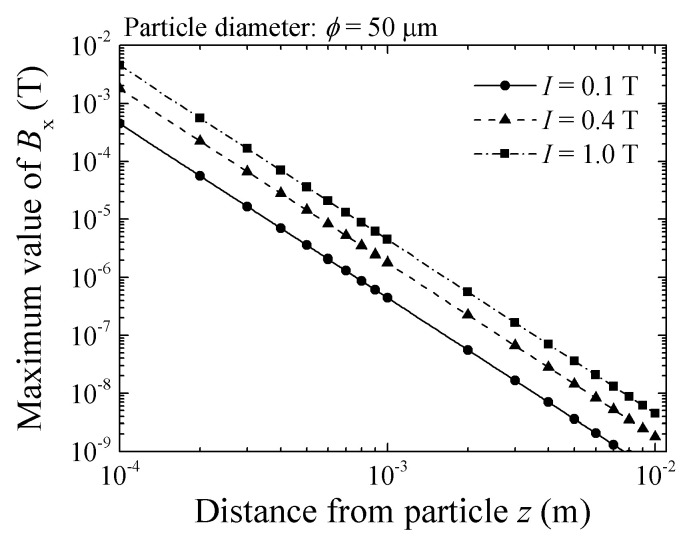
Simulated variation of flux density, *B_x_*, as a function of particle height, *z*, in case of particle diameter 50 μm [21].

**Figure 5 micromachines-13-01199-f005:**
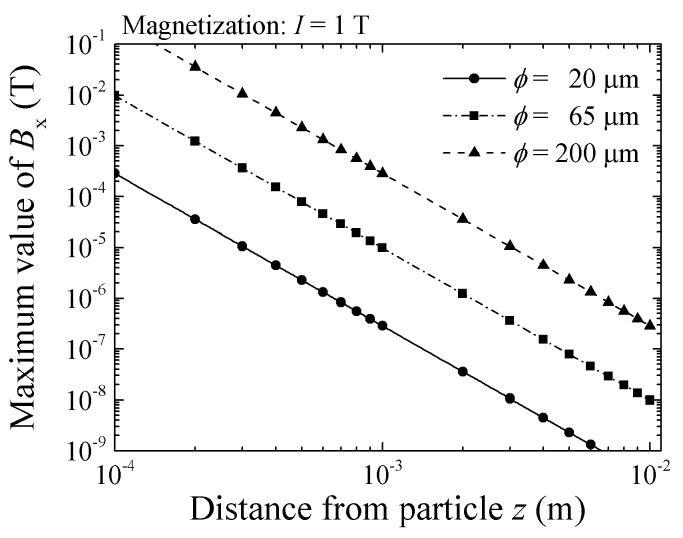
Simulated variation of flux density, *B_x_*, as a function of particle height, *z*, in case of particle magnetization 1.0 T [21].

**Figure 6 micromachines-13-01199-f006:**
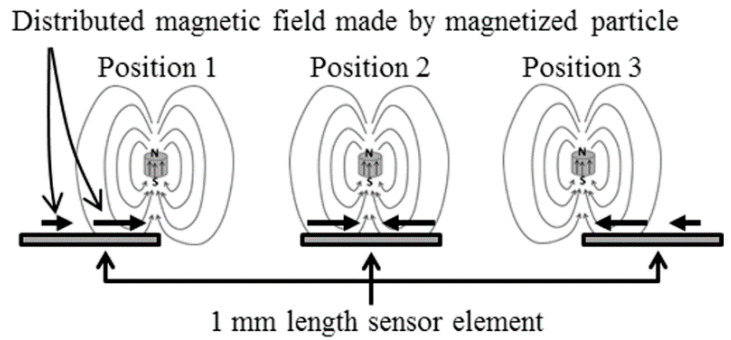
Explanation of interaction between the magnetized particle and thin film sensor in a vertical magnetic field [21].

**Figure 7 micromachines-13-01199-f007:**
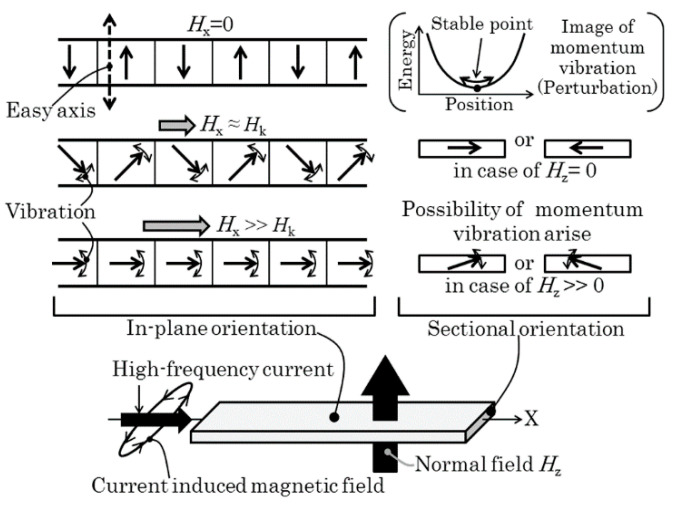
Schematic explanation of magnetization vibration in the thin film magneto-impedance sensor as a function of sensing field in X-direction, when a surface normal field is applied [21].

**Figure 8 micromachines-13-01199-f008:**
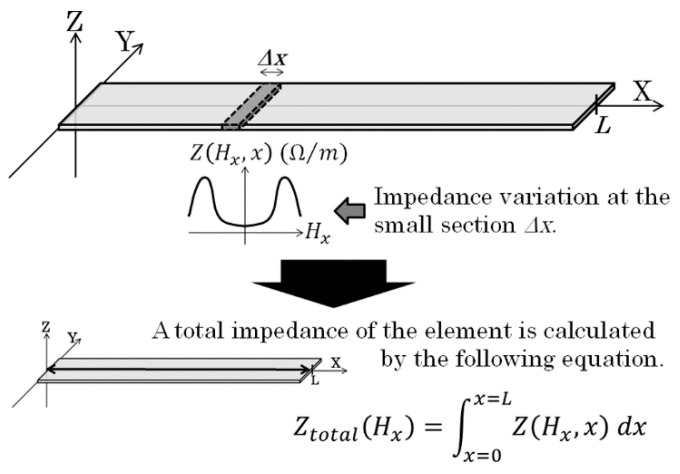
Schematic explanation of integral formula for the effect of field distribution on sensor impedance [21].

**Figure 9 micromachines-13-01199-f009:**
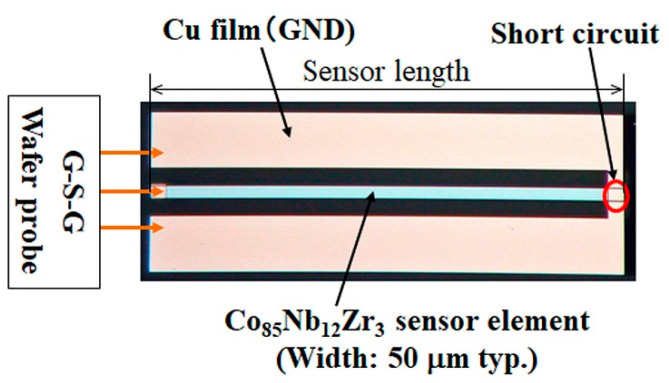
Photograph of sensor element, used for the first experiment.

**Figure 10 micromachines-13-01199-f010:**
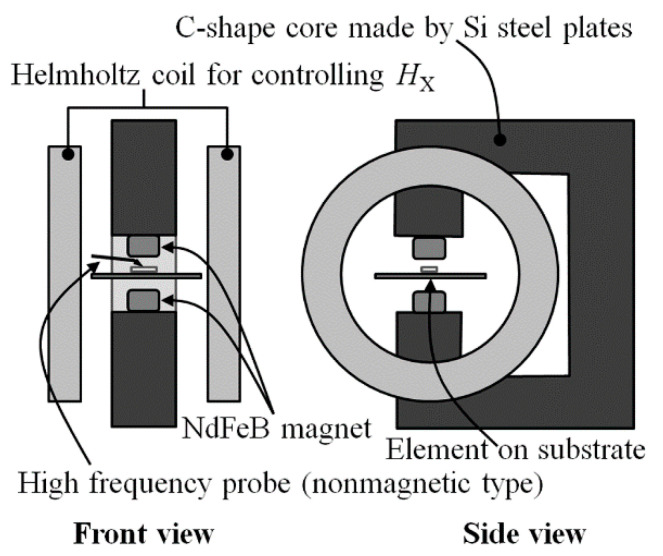
Schematic illustration of measurement apparatus [21].

**Figure 11 micromachines-13-01199-f011:**
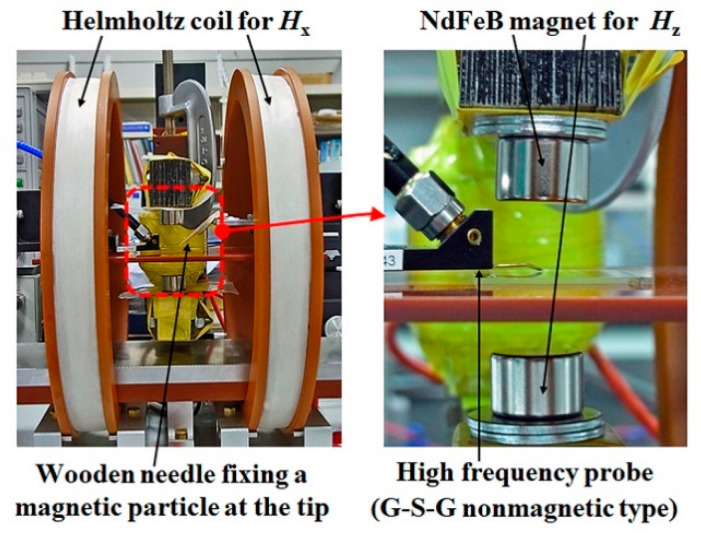
Photograph of measurement apparatus.

**Figure 12 micromachines-13-01199-f012:**
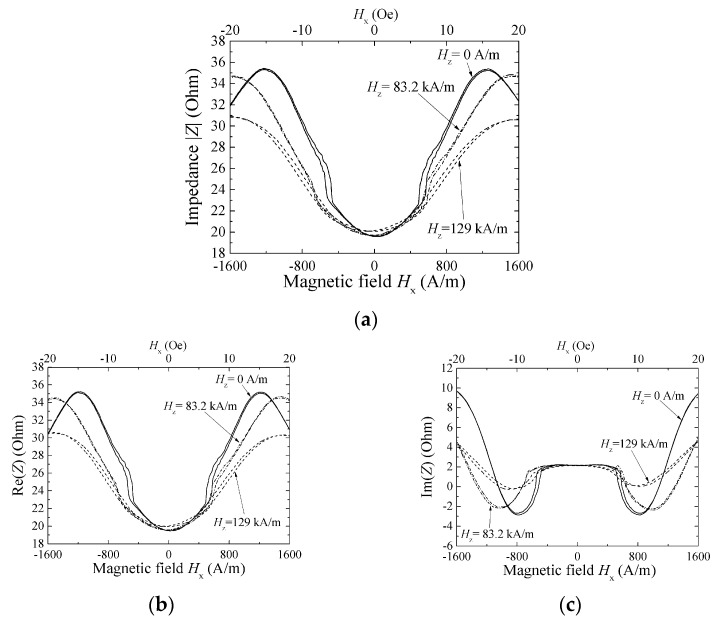
Variation of sensor impedance as a function of external field in X direction. Element dimensions, length: 1000 μm, width: 50 μm. Measurement frequency: 800 MHz. (**a**) |Z|, (**b**) Re(Z), and (**c**) Im(Z) [21].

**Figure 13 micromachines-13-01199-f013:**
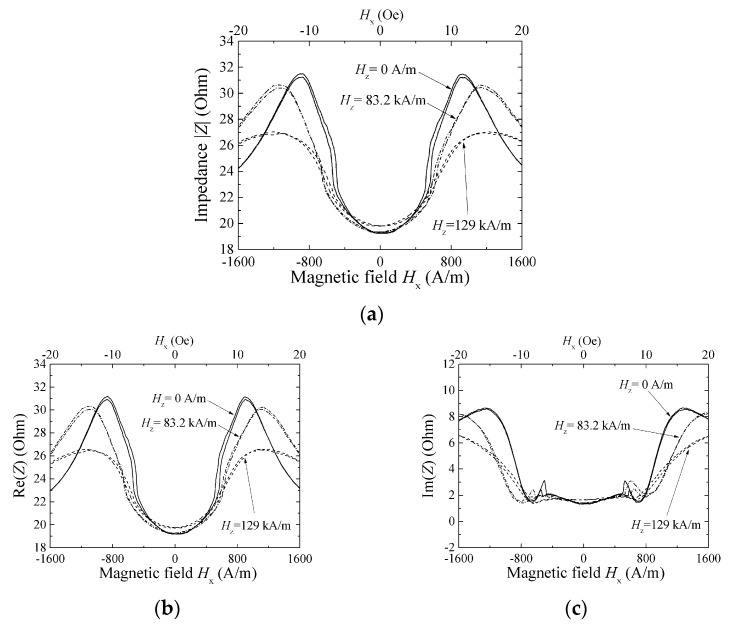
Variation of sensor impedance as a function of external field in X direction. Element dimensions, length: 1000 μm, width: 50 μm. Measurement frequency: 500 MHz. (**a**) |Z|, (**b**) Re(Z), and (**c**) Im(Z) [21].

**Figure 14 micromachines-13-01199-f014:**
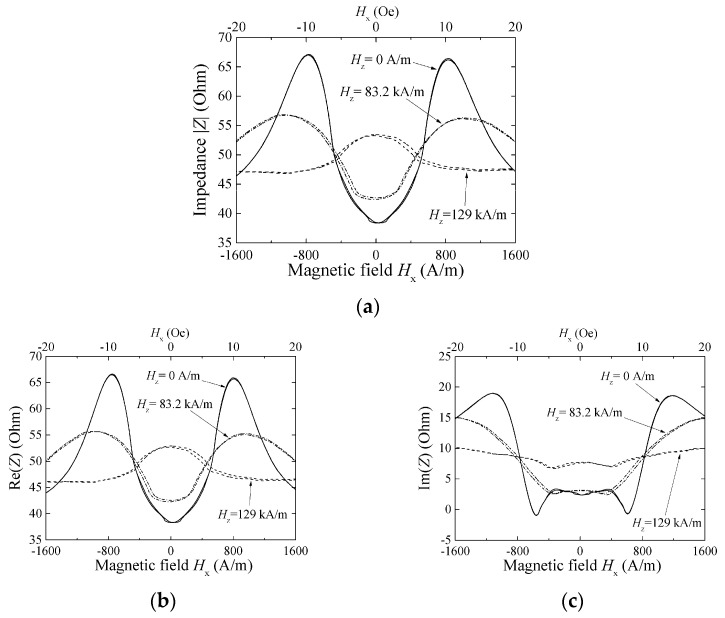
Variation of sensor impedance as a function of external field in X direction. Element dimensions, length: 2000 μm, width: 50 μm. Measurement frequency: 500 MHz. (**a**) |Z|, (**b**) Re(Z), and (**c**) Im(Z) [21].

**Figure 15 micromachines-13-01199-f015:**
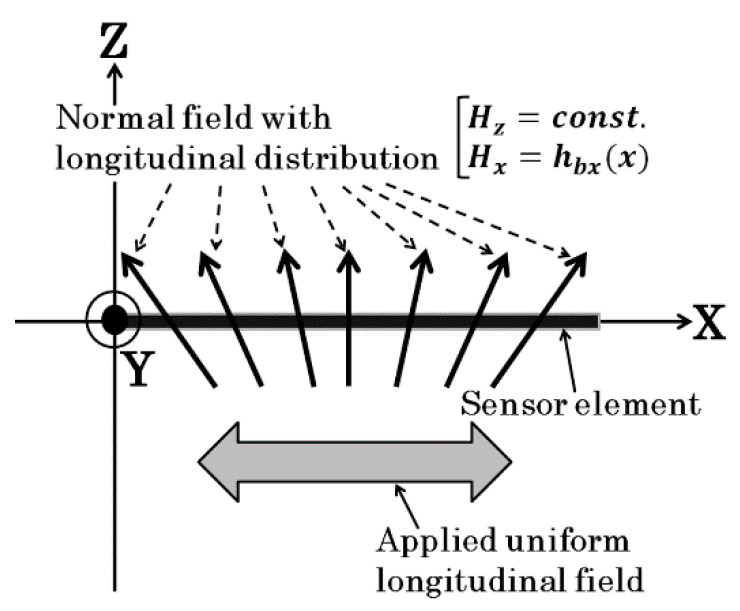
Schematic explanation of a distributed vertical magnetic field [21].

**Figure 16 micromachines-13-01199-f016:**
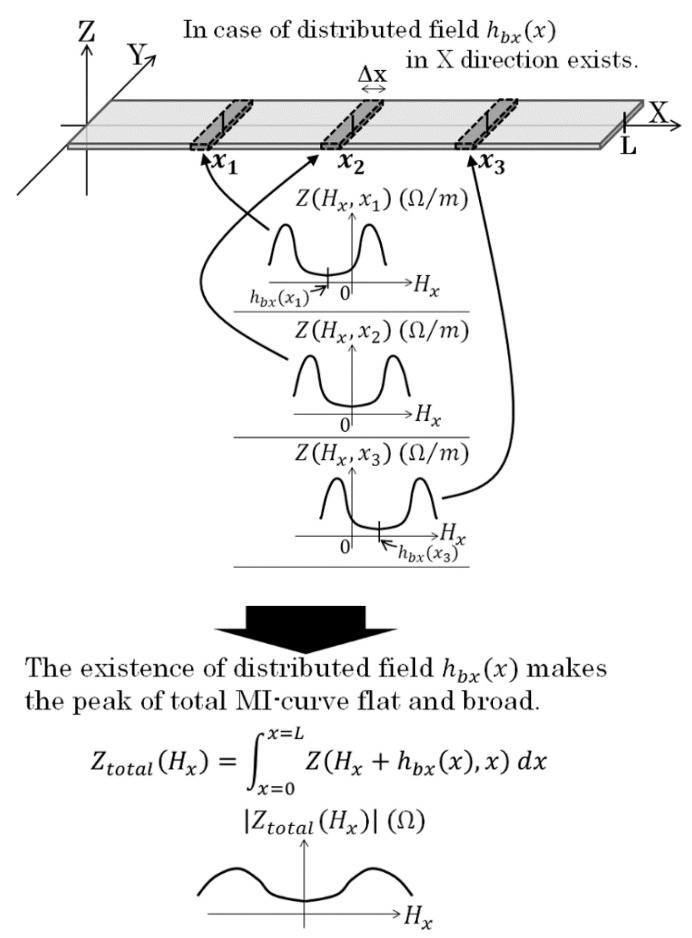
Schematic explanation of an effect of the distributed field on impedance integral [21].

**Figure 17 micromachines-13-01199-f017:**
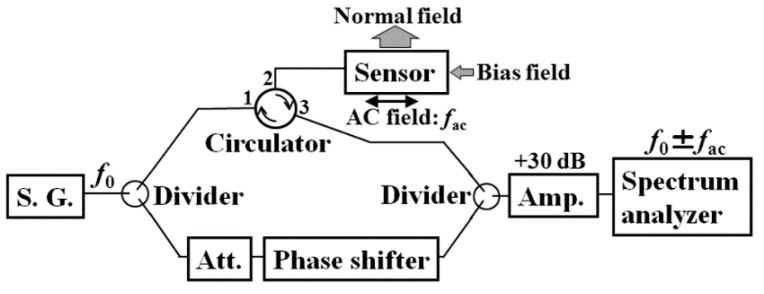
Block diagram of the carrier-suppressing circuit using reflection signal [21].

**Figure 18 micromachines-13-01199-f018:**
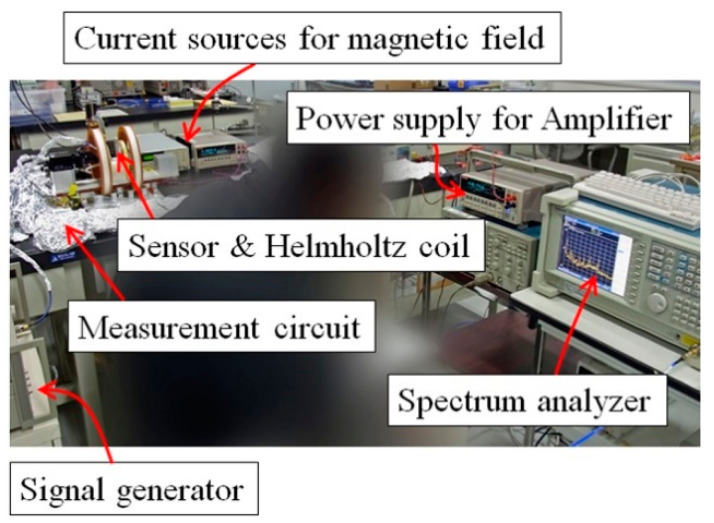
Photograph of whole measurement system.

**Figure 19 micromachines-13-01199-f019:**
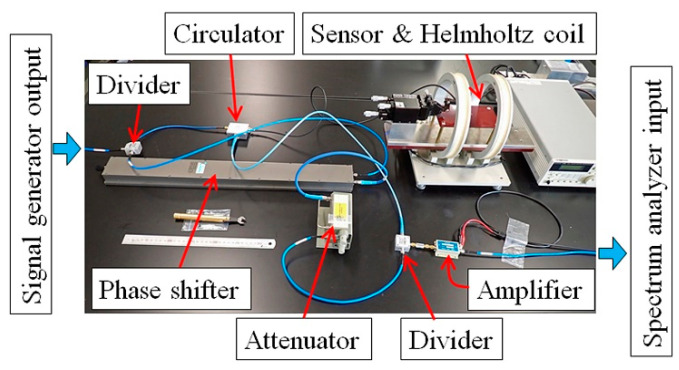
Enlarged view of the carrier-suppressing circuit connected with the sensor.

**Figure 20 micromachines-13-01199-f020:**
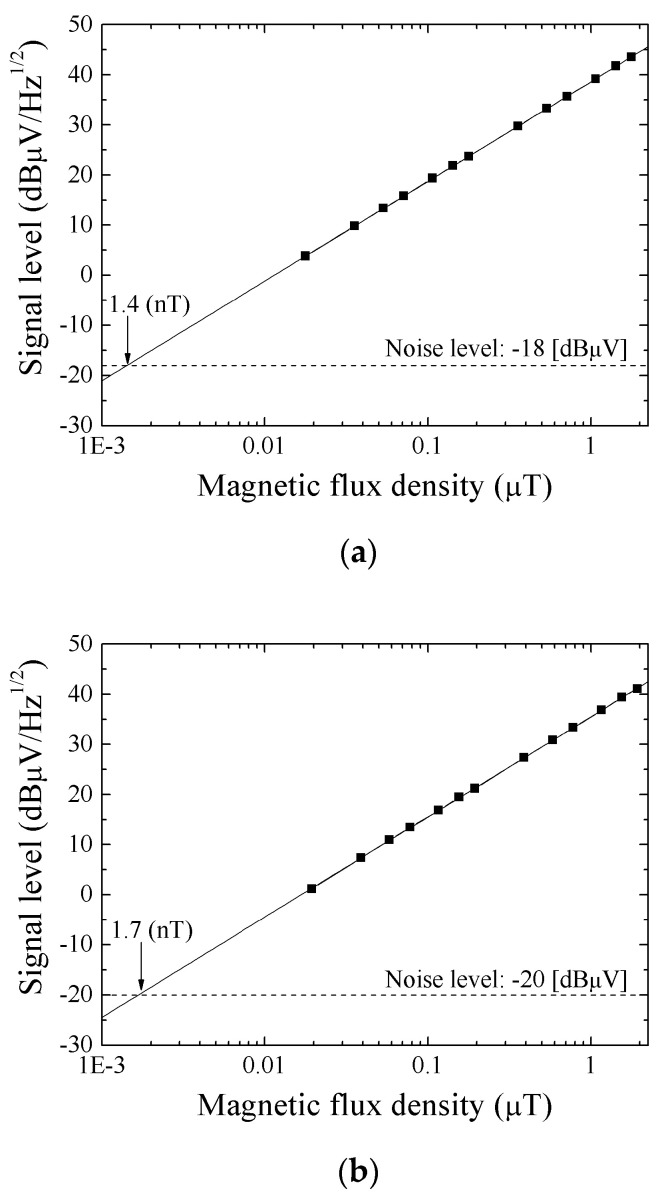
Results of evaluating sensitivity, (**a**) The surface normal field is zero, and (**b**) The surface normal field is 83.2 kA/m [21].

**Figure 21 micromachines-13-01199-f021:**
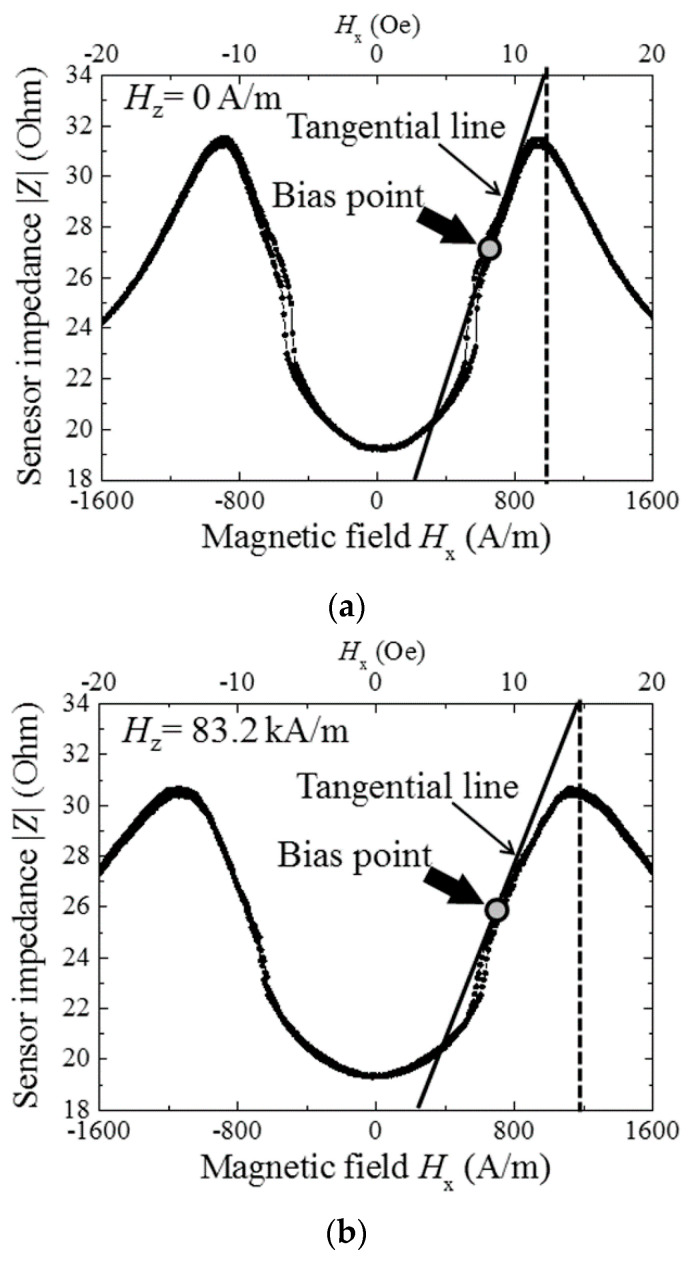
MI-curve and tangential line at bias-point of sensor operation, (**a**) The normal field is zero, and (**b**) The normal field is 83.2 kA/m [21].

**Figure 22 micromachines-13-01199-f022:**
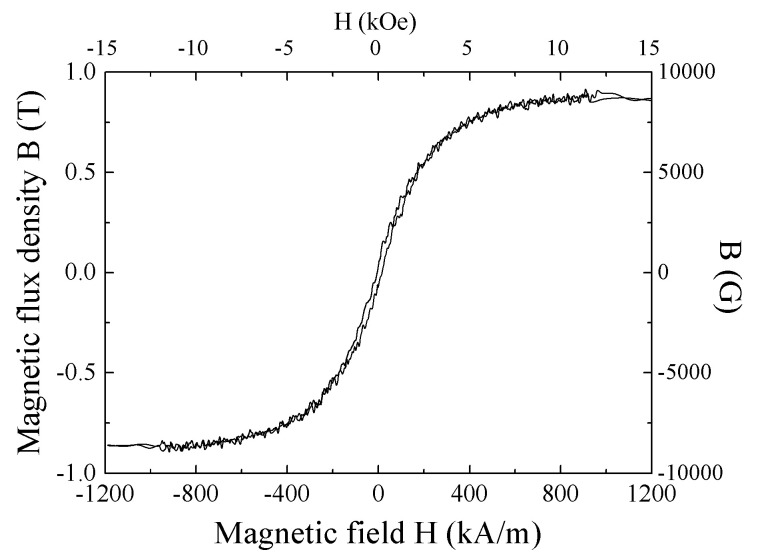
Measured M-H loop of *φ*200 μm diameter soft magnetic particle [21].

**Figure 23 micromachines-13-01199-f023:**
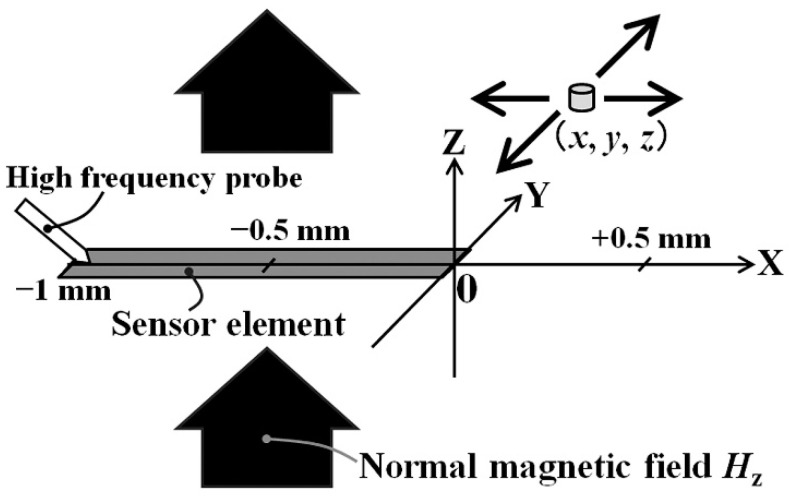
Schematic illustration of the measurement system of magnetic small particle [21].

**Figure 24 micromachines-13-01199-f024:**
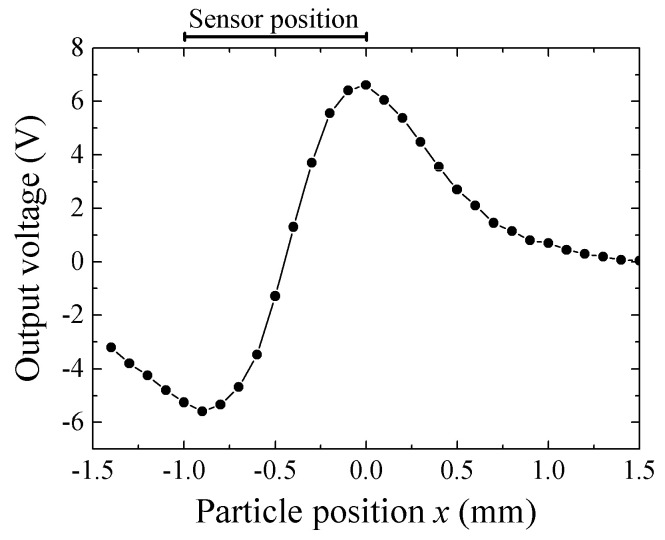
Result of measurement of scanned magnetic small particle along sensor longitudinal direction [21].

**Figure 25 micromachines-13-01199-f025:**
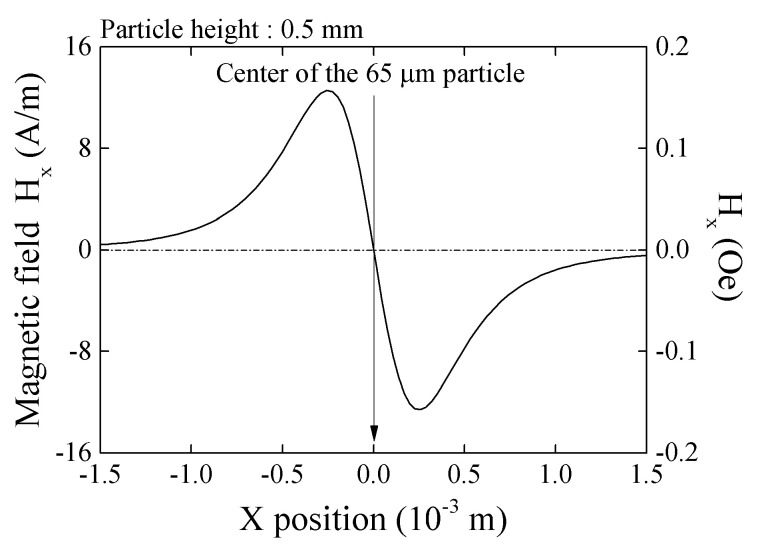
Calculated X-directional field on *X*-axis made by 0.2 T magnetized *φ*65 μm particle, placed on z = 0.5 mm [21].

**Figure 26 micromachines-13-01199-f026:**
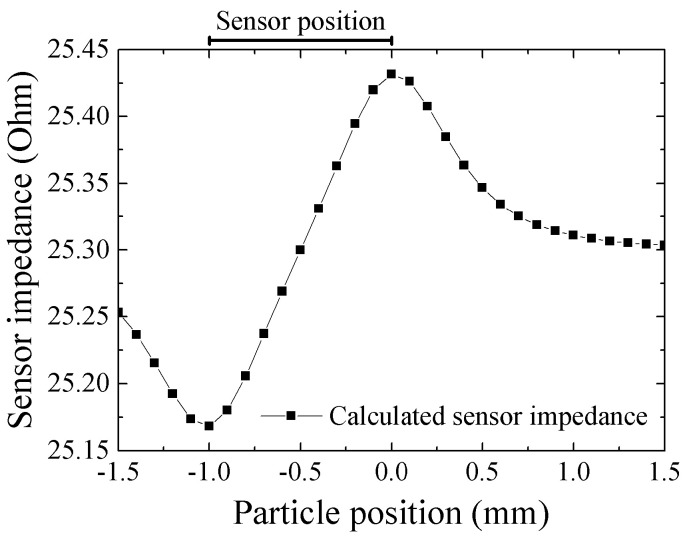
Result of numerical estimation of sensor impedance as a function of particle position [21].

**Figure 27 micromachines-13-01199-f027:**
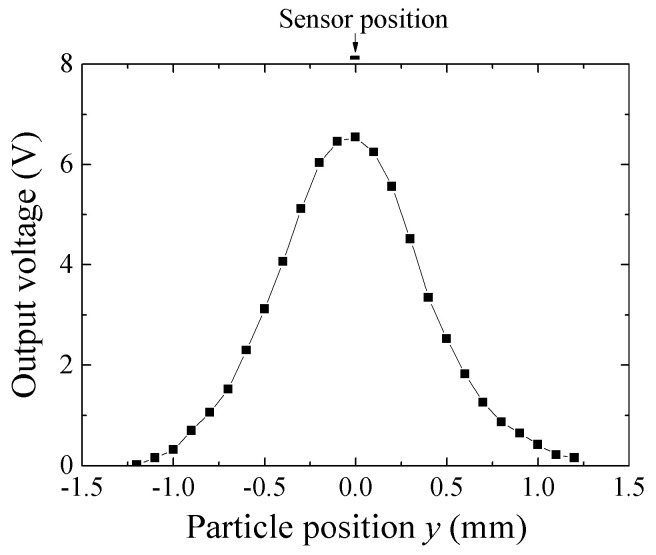
Result of experimental measurement of scanned magnetic small particle along transverse direction, *y*, on *x* = 0 [21].

**Figure 28 micromachines-13-01199-f028:**
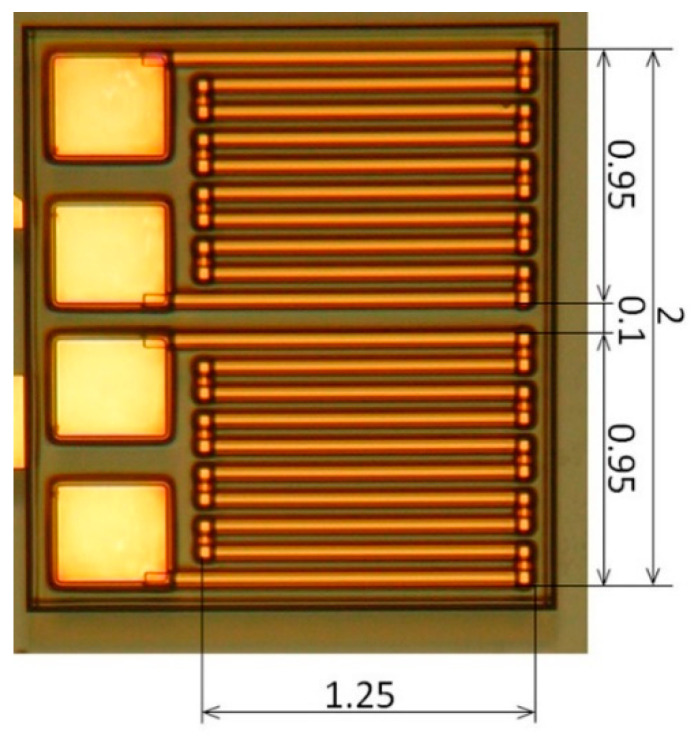
Photograph of the meander shaped differential sensor.

**Figure 29 micromachines-13-01199-f029:**
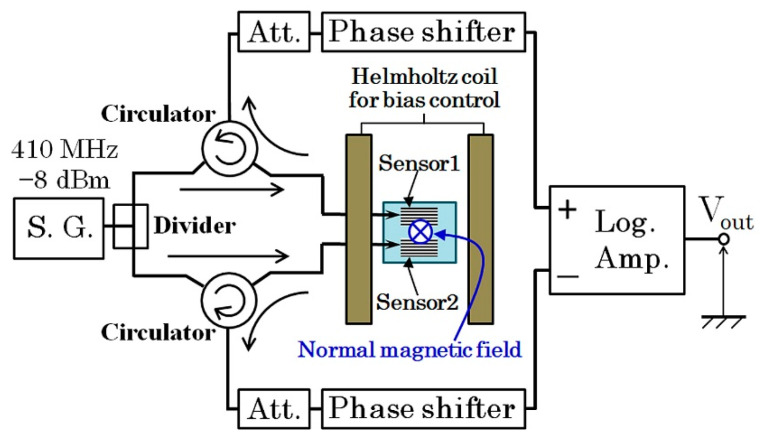
Block diagram of the driving circuit of differential sensor.

**Figure 30 micromachines-13-01199-f030:**
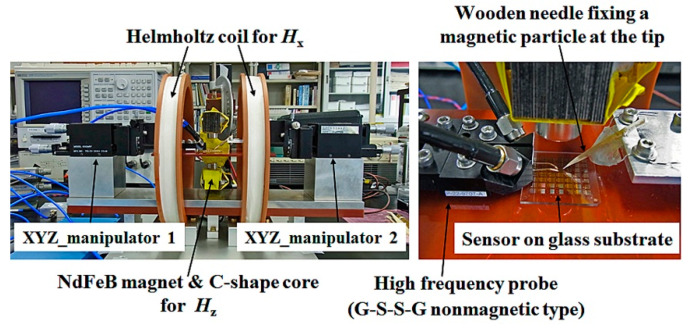
Photograph of measurement system of the differential detection experiment.

**Figure 31 micromachines-13-01199-f031:**
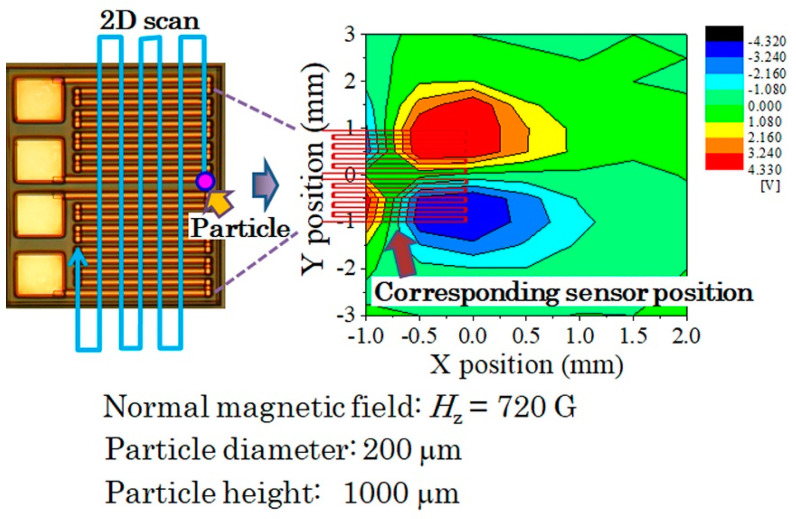
2D-mapping of a measured signal, when a particle having a diameter of 200 μm was scanned on the sensor element with the height of 1 mm.

**Figure 32 micromachines-13-01199-f032:**
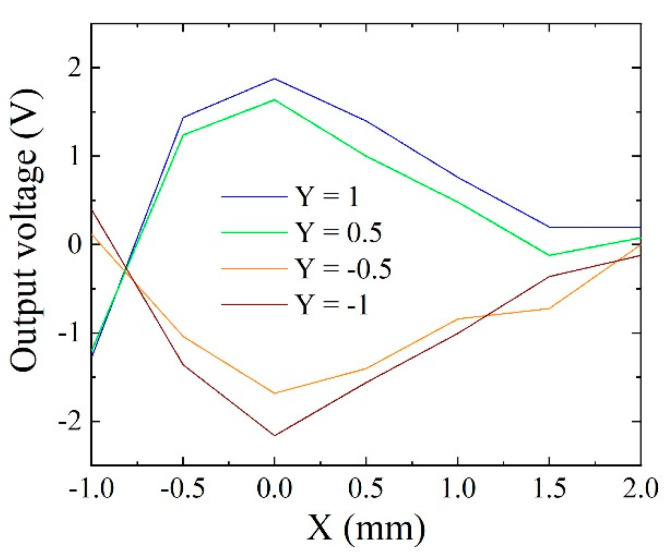
Output profiles as a function of X-position, when a particle having a diameter of 200 μm was scanned on the sensor element with the height of 1.5 mm.

**Table 1 micromachines-13-01199-t001:** Device list of the circuit.

	Function	Model Number	Manufacturer
1	Signal generator	MG3633A	Anritsu Corporation (5-1-1 Onna, Atsugi-shi, Kanagawa 243-8555, JAPAN)
2	Divider	(Conventional 3-port resistive divider (50))	
3	Circulator	ADC040CSWH	ADMOTECH Inc. (1320, Gwanpyung-Dong, Yusong-Gu, Daejon 305-509, SOUTH KOREA)
4	Attenuator	Combination of 8594A and 8595A	Keysight Technologies (1400 Fountaingrove Parkway, Santa Rosa, CA 95403 USA)
5	Phase sifter	P1103	Narda-ATM Inc. (435 Moreland Rd, Hauppauge, NY 11788 USA)
6	Spectrum analyzer	RSA3408A	Tektronix (14200 SW Karl Braun Dr, Beaverton, OR 97077 USA)

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
