# Peer review of "Study on Detection of a Small Magnetic Particle Using Thin Film Magneto-Impedance Sensor with Subjecting to Strong Normal Field"

_micromachines, 2022, doi:10.3390/mi13081199_

Round 1

Reviewer 1 Report

First of all, the research is a greatly modernized version of the detection of very low magnetism for thin-film magneto-impedance technology. People assume that this technology has many positive aspects. In this respect, it is clear that the publication of the study in a prestigious scientific journal will inspire other researchers in this field. As with any research, there will be flaws in this study, and these flaws are detailed below.

1- However, an in-depth reading reveals that the motivation element of the study is not explained in depth enough. The visibility and quality of work is not expected to benefit from rethinking and rewriting the motivation part, as some suggestions for doing this below show. The references given below, which are thought to contribute to the study, are recently directed to both thin film and general magnetoimpedance studies.

            https://doi.org/10.1002/tee.23186

            https://doi.org/10.1134/S0031918X19130143

            https://doi.org/10.1007/s00339-019-3096-5

            https://doi.org/10.1016/j.jmmm.2021.168356

            https://doi.org/10.3390/s21186145

2- Another concern is that the article's findings aren't fully covered in other sources of literature. Doing so is quite crucial.

3- Although the part mentioned in the work as carrier-suppressing circuit is detailed, it would be more appropriate to have a real picture.

4- Finally, according to the data given in figure 30, why couldn't a symmetrical result be obtained in terms of output voltage exactly according to the change in position of outer end strip of the meander sensor (Y). This result makes it difficult to compare the results obtained at different Y values.

Reviewer 2 Report

The work discuss the physical principles and experimental validation of small magnetized particle detection using strong vertical magnetic field. In general, the variation of sensor impedance coming from the magnetic particles was estimated taking into account both long and short distance to the sensor element. The work is solid from both from the theoretical as well as the technical/experimental point of view. The manuscript is written using good, and professional English. I do not find any weak points of this paper, therefore I recommend to accept it for publication in the current form.

Reviewer 3 Report

The presented study deals with the detection of small magnetization using a thin film magneto-impedance sensor made from soft magnetic amorphous thin-film, when subjecting to strong normal field. The author proposes a sensitive measurement method aiming for detection of a small particle or a cluster of nano-particles, having low-remanence. The author reports a theoretical estimation for clarifying an efficiency of the proposed method, experimental results of sensor property and sensitivity with subjecting to the strong normal field, as well as a confirmation of detection of a small particle using this method. It has to be mentioned that this sensor has a possibility for detecting a certain kind of nonmagnetic conductive materials, in addition of the detection of ordinal magnetic materials. The author proposes a method which combines a high-sensitivity sensor made by thin film and application of strong magnetic field in the measurement area. The strong magnetic field is applied for the purpose of magnetize the low-remanence particle. Instead of such strong normal field, the thin film sensor keeps high sensitivity, because of the demagnetizing force in the thickness direction.

The method and results presented in this manuscript are original and interesting for the readers of this journal. I  suggest to accept after minor revision.

Comments:

1) Usually, the abbreviation of magnetization is M. The author uses I. Of course, it is the decision of author to change or not, because the I is defined in the text.

2) The authors present some data of magnetization depending on the position and particle diameter (for example Fig.3, 4, etc.). Please provide more clear explanation how the magnetization was evaluated depending on particle size. Also: did you measure the magnetization or just the signal of the sensor?

Reviewer 4 Report

Magnetoimpedance detecting of magnetic particles are relevant in various applications, especially in biomedicine. Practice puts forward special requirements for the sensitivity of such sensors associated with an extremely small value of the detected field. The author proposes and discusses the design of a sensor based on a magnetic impedance sensor coupled to a source of a sufficiently strong magnetic field. This idea seems to be fruitful, since the presence of a constant magnetizing field increases the signal of the detected particles. The author gives the necessary theoretical calculations, discusses the details of the sensor design, and gives primary data on particle detection. In my opinion, the article in general is worthy of publication. Let me make one remark.

It would be desirable to see the comparison of the main parameters of the proposed sensor with detectors of other configurations, in particular, without the use of a constant magnetizing field. Perhaps the author should look at the latest review on this issue [Jimenez, V.O.; Hwang, K.Y.; Nguyen, D.; Rahman, Y.; Albrecht, C.; Senator, B.; Thiabgoh, O.; Devkota, J.; Bui, V.D.A.; Lam, D.S.; et al. Magnetoimpedance Biosensors and Real-Time Healthcare Monitors: Progress, Opportunities, and Challenges. Biosensors 2022, 12, 517, doi:10.3390/bios12070517. ], as well as some articles related to the design of impedance sensors [Kurlyandskaya, G.V.; Sánchez, M. L.; Hernando, B.; Prida, V.M.; Gorria, P.; Tejedor, M. Giant-Magnetoimpedance-Based Sensitive Element as a Model for Biosensors. Appl. Phys. Lett. 2003, 82, 3053-3055.; Kurlyandskaya, G. V.; Levit, V. Magnetic Dynabeads Detection by Sensitive Element Based on Giant Magnetoimpedance. Biosens. Bioelectron. 2005, 20, 1611-1616.; Kurlyandskaya, G. V.; Fal Miyar, V. Surface Modified Amorphous Ribbon Based Magnetoimpedance Biosensor. Biosens. Bioelectron. 2007, 22, 2341–2345].
